# Effect of Music Intervention on Lung Expansion Exercises after Cardiothoracic Surgery

**DOI:** 10.3390/jcm11061589

**Published:** 2022-03-13

**Authors:** Guan-Yi Chen, Lan-Yuen Guo, I-Chun Chuang, Ho-Chang Kuo, Yuh-Chyn Tsai, Shih-Feng Liu

**Affiliations:** 1Department of Respiratory Therapy, Kaohsiung Chang Gung Memorial Hospital, Kaohsiung 833, Taiwan; millenniayi@yahoo.com.tw (G.-Y.C.); erickuo48@yahoo.com.tw (H.-C.K.); jane2793@cgmh.org.tw (Y.-C.T.); 2Department of Sports Medicine, College of Medicine, Kaohsiung Medical University, Kaohsiung 807, Taiwan; yuen@kmu.edu.tw; 3Department of Medical Research, Kaohsiung Medical University Hospital, Kaohsiung 807, Taiwan; 4College of Humanities and Social Sciences, National Pingtung University of Science and Technology, Pingtung 912, Taiwan; 5Department of Respiratory Therapy, Kaohsiung Medical University, Kaohsiung 807, Taiwan; icchuang@kmu.edu.tw; 6Department of Paediatrics, Kaohsiung Chang Gung Memorial Hospital, Kaohsiung 833, Taiwan; 7Medical Department, College of Medicine, Chang Gung University, Taoyuan 333, Taiwan; 8Division of Pulmonary and Critical Care Medicine, Department of Internal Medicine, Kaohsiung Chang Gung Memorial Hospital, Kaohsiung 833, Taiwan

**Keywords:** music intervention, cardiothoracic surgery, lung function, pain, anxiety

## Abstract

Background: Music intervention can reduce anxiety. This study analyzed the physiological changes from using music intervention after cardiothoracic surgery. Methods: Subjects were randomly assigned to the music group or the control group. The maximal inspiratory pressure/maximal expiratory pressure (MIP/MEP), pulse oximeter oxygen saturation (SpO2), visual analogue scale (VAS) for pain, and State-Trait Anxiety Inventory (STAI) were compared. Results: Compared to the control group (*n* = 9), the music group (*n* = 9) had higher MIP and MEP during the overall test (*p* < 0.05), with significant differences in the changes and time (*p* < 0.001). However, only MEP was significant in terms of the interaction between music intervention and time (*p* < 0.001). In terms of the groups, SpO2 and VAS were significant (*p* < 0.05). SBP, SpO2, and VAS over time showed significant differences between the two groups (*p* < 0.05). In terms of the interaction between music intervention and time, only SpO2 was significant (*p* < 0.05). The STAI-S scale decreased by −5.7 ± 5.8 in the music group vs. −0.47 ± 9.37 in control group and the STAI-T scale increased by 4.17 ± 12.31 in the music group vs. 1.9 ± 9.29 in the control group, but showed no significance. Conclusions: Music intervention with nature sounds has a positive physiological impact and can reduce postoperative pain and anxiety in cardiothoracic surgery patients.

## 1. Introduction

Most people have to undergo surgery or invasive tests or treatments at some point in their lives. Music intervention can be used to relieve preoperative anxiety, intraoperative restlessness, or postoperative pain. Past studies have pointed out that the use of music interventions in patients waiting for cardiothoracic surgery can significantly reduce anxiety [1]. Providing music to postoperative patients can help the patient feel relaxed and more at home in the hospital.

After cardiothoracic surgery, patients cannot immediately leave bed or walk around, and the pulmonary recovery they receive mostly comprises intermittent positive pressure breathing (IPPB), which can increase peak expiratory flow and cough efficacy [2], as well as reduce the time required for postoperative recovery and incidences of pneumonia.

Before a patient undergoes surgery, a respiratory therapist provides training on IPPB; most patients respond positively. After surgery, however, some patients may suffer from pain and, therefore, display anxiety or nervousness or may not want to cooperate with the performance of these exercises. In addition, patients who are willing to cooperate might only perform the exercises for a short time, which leads to poor results. Both of these scenarios may lead to pneumonia due to insufficient lung expansion, which increases the likelihood of requiring intubation and even increases the mortality rate.

Therefore, this study aimed to investigate postoperative lung expansion exercises in patients undergoing cardiothoracic surgery and to compare the effectiveness of intermittent positive pressure breathing (IPPB) with and without music intervention.

## 2. Materials and Methods

### 2.1. Study Design

This study was a randomized controlled trial, in which subjects were randomly assigned to receive music intervention (music group) or no music intervention (control group). Then, their related pulmonary function and anxiety scales were evaluated, and their clinical vital signs were recorded. Finally, the benefits of music in lung expansion exercises were analyzed. This experiment was reviewed and approved by the Chang Gung Medical Foundation Institutional Review Board, IRB case number: 201901501B0.

### 2.2. Study Participants

The enrollment criterion was adults aged 20 years or older who had undergone cardiothoracic surgery. The subject or their family signed a consent form. The exclusion criteria were as follows: 1. postoperatively unconscious or uncooperative; 2. hemodynamic instability; 3. untreated pneumothorax; 4. severe infection; 5. chronic obstructive pulmonary disease; 6. esophagotracheal fistula; 7. pulmonary tuberculosis; 8. severe systemic diseases such as end-stage cancer, end-stage renal disease, liver cirrhosis, or congestive heart failure. No past studies have examined patients’ lung function after music intervention; instead, they mostly evaluated changes to lung function in COPD patients after music intervention, and most of them showed no significant differences.

### 2.3. Randomization

A total of 22 blank CDs were prepared. Nature music was burned onto 11 CDs, which were used by the music group. The other 11 CDs were silent, which were used by the control group. A burned CD has noticeable etching marks on the read side. In order to maintain the blindness of the experiment, both sets of CDs were burned for 15 min so that the etchings on the read surface of the CDs used in both groups were the same. The burnt CDs and the subjects’ informed consent forms were placed in opaque sealed envelopes, and a member of the study’s staff was asked to shuffle the envelopes. Subjects in both groups wore SONY WH-CH700N (Sony, Tokyo, Japan) noise-canceling headphones during the experiment and listened to the CDs during lung expansion therapy only. Disposable headphone covers were used in this study to prevent infections and ensure sanitation. Since music preferences can vary widely between people and to prevent experimental errors, the music selected comprised nature sounds that are broadly accepted [3]. The music selection was “Nature Sounds: Forest Sounds, Birds Singing and Sound of Water” (https://youtu.be/d0tU18Ybcvk, accessed on 24 February 2020).

### 2.4. Data Collection

For the primary outcome of the study, the maximal inspiratory pressure (MIP) and maximal expiratory pressure (MEP) were measured with a Boehringer^™^ pressure gauge, and lung capacity was measured with a handheld respirometer, Haloscale^®^ (nSpire Health™, Longmont, CO, United States). A peak expiratory flow meter (Trudell Healthcare Solutions Inc., London, Canada) was used to measure the peak expiratory flow rate. The secondary outcome was a visual analogue scale (VAS) used to assess the level of pain, in which 0 means no pain and 10 means a tremendous amount of pain. The state-trait anxiety scale was used to measure the presence and severity of anxiety symptoms and general anxiety tendencies in the patient before surgery and on the seventh day of lung expansion exercises. A physiological monitoring system (IntelliVue MP60, Philips Medizin Systeme Böblingen GmbH, Böblingen, Germany) was used to monitor the patient’s vital signs and recorded the heartbeat (HR), systolic and diastolic blood pressure (SBP and DBP), and blood oxygen saturation (SpO_2_).

### 2.5. Statistics

The study subjects were divided into either the music group or the control group, and the duration of the study was seven days. The data obtained were analyzed via 2-way ANOVA, with an α value of 0.05 and a β value of 0.8. Assuming that the partial eta-squared (η^2^) was 0.06, a medium effect could be achieved. The total subject population to be calculated using G*Power software was 18 subjects, but after taking into account a 20% attrition rate, an estimated 22 people were accepted into this study [4]. The Shapiro–Wilk test was used to test the subjects for a normal distribution. For the music intervention, a generalized estimating equation (GEE) was used to obtain the regression equations for measurements of clinical vital signs, related pulmonary physiological functions, and the degree of pain. Partial eta-square (η^2^) was used to represent the effect size. For the differences in the State-Trait Anxiety Inventory (STAI) before and after the trial, an independent *t*-test was used to test the effect of music intervention on the two groups. Kendall’s τ correlation was used to test the correlations of the average variations of each parameter. The software used was IBM SPSS Version 23 (IBM Corp., Armonk, NY, United States).

## 3. Results

### 3.1. Differences between the Normal Distribution and the Two Groups before the Trial

The number of pre-enrollment subjects in this trial was 22. Among them, 2 refused to participate in the trial, and 2 did not undergo sternotomy, so 18 subjects were ultimately included in the analysis. All patients underwent sternotomy, and 18 cases were operated by type, including 7 cases of coronary artery bypass grafting, 6 cases of valve replacement, 3 cases of aortic arch replacement, 1 case of atrial septal defect repair, and 1 case of tumor resection (Table 1).The systolic blood pressure of the music group and the status of the anxiety scale for the control group did not conform to a normal distribution. A nonparametric Wilcoxon signed-rank test was used to determine that there were no significant differences in the pre-trial baseline values of the music group and control group (Table 2).

### 3.2. Music of Model Effects in Pulmonary Function in the Music Group and the Control Group

There were no significant differences between the music group and the control group in the overall test process; only the effect sizes of the maximal inspiratory pressure and the maximal expiratory pressure reached a medium effect or higher. There were statistically significant differences between the music group and the control group in the changes in lung function and time (Figure 1). In terms of the interaction between music intervention and time, only the maximal expiratory pressure showed no significant difference (Table 3).

### 3.3. Effects of Music on Vital Signs and Visual Analogue Scale for Pain in the Music Group and the Control Group

The effects of music on vital signs and the visual analogue scale for pain were tested using the same statistical methods, and the results of the analysis are listed in Table 4. Only the blood oxygen saturation and visual analogue scale for pain were significantly different between the two groups (*p* < 0.05). In terms of the effect size, heartbeat, blood oxygen saturation, and visual analogue scale for pain (Figure 2), all achieved medium effects or above. There were statistically significant differences (*p* < 0.05) between the music group and the control group in the changes over time in systolic blood pressure, oxygen saturation, and visual analogue scale for pain. In terms of the interaction between music intervention and time, only blood oxygen saturation showed a statistically significant difference (*p* < 0.05; Table 4 and Figure 3).

The effect of music on the visual analogue scale for pain was analyzed for both the music group and the control group, and a statistically significant difference was found (*p* < 0.05), indicating that the music group’s perception of pain was affected by the music intervention from the beginning of the trial. The results of the study showed that the perception of pain changed daily with time, and the effect was statistically significant (*p* < 0.05). Furthermore, Figure 2 shows that there was a significant difference in the visual analogue scale for pain between the two groups on the seventh day after treatment (*p* = 0.016), and the VAS score of the music group was lower than that of the control group.

### 3.4. Changes of State-Trait Anxiety Scale before and after Surgery in the Control Group and the Music Group

The STAI scales before surgery and seven days after interventional lung expansion exercises with and without music were compared. In terms of the average change in the test, the post-test value was subtracted by the pre-test value, and the music group showed a decrease of −5.667 ± 5.729 points on the STAI-S scale, while the control group showed a decrease of −0.444 ± 9.302 points (*p* = 0.171). On the STAI-T scale, the mean changes in the music group and the control group were both increases, in which the music group increased by 4.111 ± 12.313 points, while the control group increased by 1.889 ± 9.226 points (*p* = 0.671). There were no statistically significant differences between the preoperative and postoperative days of the anxiety state (STAI-S) scale or the anxiety trait (STAI-T) scale (*p* > 0.05; Table 5 and Figure 4).

## 4. Discussion

The average vital capacity value in a normal adult is approximately 50 mL/kg, while the average value for the subjects in this study was 25.79 mL/kg, or approximately one-half of the normal value. This indicates that the inspiratory or expiratory muscles were weak. The symptoms of the acute phase of heart disease are mostly shortness of breath, dyspnea, and reduced lung capacity. The maximal inspiratory pressure and the maximal expiratory pressure also did not reach the normal values, but there was no effect on ventilation and cough function. The definition of no effect was: MIP ≤ 60 cmH_2_O and MEP > 60 cmH_2_O [5,6]. The peak expiratory flow rate can indicate the patient’s level of cough function and whether there are airway obstructions. The obtained values mostly fell between the yellow zone (attention) and the green zone (safe). The measurement of pulmonary physiological functions are all voluntary maneuvers, and the main measurements are closely related to the patient’s compliance [7]. To explore the relationship between the respiratory muscle (diaphragm) and heart disease, Kelley and Ferreira proposed that factors such as higher age or heart disease in patients increase the release of reactive oxygen species (ROS) in the body, inducing increased activity of calpain and leading to skeletal muscle fatigue and weakness [8].

To obtain the vital capacity value, the patient was asked to take a deep breath and then exhale hard until there was no more air left. Mohan et al. pointed out that forced exhalation is mainly achieved through the contraction of the intercostal muscles [9]. The peak expiratory flow rate represents the patient’s cough function and airway stenosis. Ishida et al., in an electromyography study, found that the peak expiratory flow rate depends on the work of the external oblique muscle and the transverse abdominis, and requires the rapid contraction of the intercostal muscles [10].

To further examine why the peak expiratory flow rate and vital capacity measured in the control group were higher than those in the music group, Jinakote et al. found, through intercostal electromyography studies, that the increase in postoperative respiratory muscle load was related to weakened vagal control [11]. Furthermore, Strauss-Blasche et al. also confirmed that deep breathing can stimulate the vagus nerve [12], while Iwanaga et al. found that the vagus nerve has a higher level of activity when the subject listens to calming music [13]. Therefore, we inferred that the music group had higher vagal activity when they listened to music, reducing the activation potential of the intercostal muscle and resulting in a lower peak expiratory flow rate and spirometry than those measured in the control group. The degree of pain in the control group negatively correlated with the vital capacity, which may also have been related to the magnitude of the intercostal muscle’s potential. However, there is currently no suitable literature that supports this conclusion, and further research is needed.

In an animal study, De Troyer found that the intercostal muscles and the diaphragm have a synergistic effect in lung expansion. In other words, the performance of the intercostal muscles can reduce the shortening of the diaphragm [14]. Therefore, we inferred that after the intervention with music, the average values of the maximal inspiratory pressure and the maximal expiratory pressure in the music group were greater than those in the control group, which may have occurred because the activation of the intercostal muscles was reduced by music, and the performances of the diaphragm and rectus abdominis were more obvious.

For blood oxygen saturation, Cimen et al. examined the use of music intervention for surgery, and divided subjects into a control group and a music group. The music group had a higher blood oxygen saturation than that of the control group after surgery [15]. The blood pressure measurements in this trial, however, differed from those in Cimen’s study, in which there were significant differences between the control group and the music group. This may be related to the discontinuation of analgesics (e.g., dexmedetomidine, Precedex^®^), as they inhibit the release of peripheral norepinephrine and reduce pain, and norepinephrine is a neurohormone that can affect blood pressure [16]. Listening to music can also reduce the release of neurohormones, causing blood vessels to dilate and oxygen to be more effectively transported to surrounding tissues. This is why the music group had a higher blood oxygen saturation [17].

Patients may be moderately anxious prior to surgery, possibly because of insufficient sleep or fear of surgical complications after being informed about the surgical risks [18]. Patients in the control group may have blamed themselves because they realized they performed poorly in the lung functions test or because they consciously detected difficulty expectorating. Consequently, there was little difference in the anxiety state before and after surgery [19]. There is no research to explain the increase in postoperative anxiety traits. However, after the completion of this study, interviews with patients revealed that many of them were worried about family members not having anyone to take care of them or about the costs of the surgery and hospitalization.

The subjects for this trial were randomly assigned. However, in order to see the effect of lung expansion exercises after music intervention in the short term, the trial could not use a crossover trial design. Therefore, patients in the control group had lower levels of cooperation on the third or fourth day of the trial. Fortunately, they responded well after communication. For reasons of experimental consistency, the patients in the experimental group could not choose their own music. This may have affected the data collected in this experiment because musical tastes can vary between people. During the explanations of this experiment, most of the patients were worried about having to purchase additional equipment at their own expense (e.g., headphones). The patients were informed of the purpose of the experiment and that they would not be subject to additional charges. However, to prevent future disputes, some eligible subjects were not enrolled.

Future research should explore the effectiveness of lung expansion exercises with the intervention of different types of music. However, this means that more research subjects will be required in order to determine the most effective music choices in each age group for lung expansion exercises. Additionally, although we used G*Power software to calculate the number of participants, the sample size was small, but we think that the more participants, the stronger the power of the results.

## 5. Conclusions

Nature music intervention had a positive physiological impact and could reduce postoperative pain and anxiety in cardiothoracic surgery patients. To perform lung expansion early for patients who will be receiving cardiothoracic surgery, clinicians or family members can be advised to prepare music and headphones for them in advance.

## Figures and Tables

**Figure 1 jcm-11-01589-f001:**
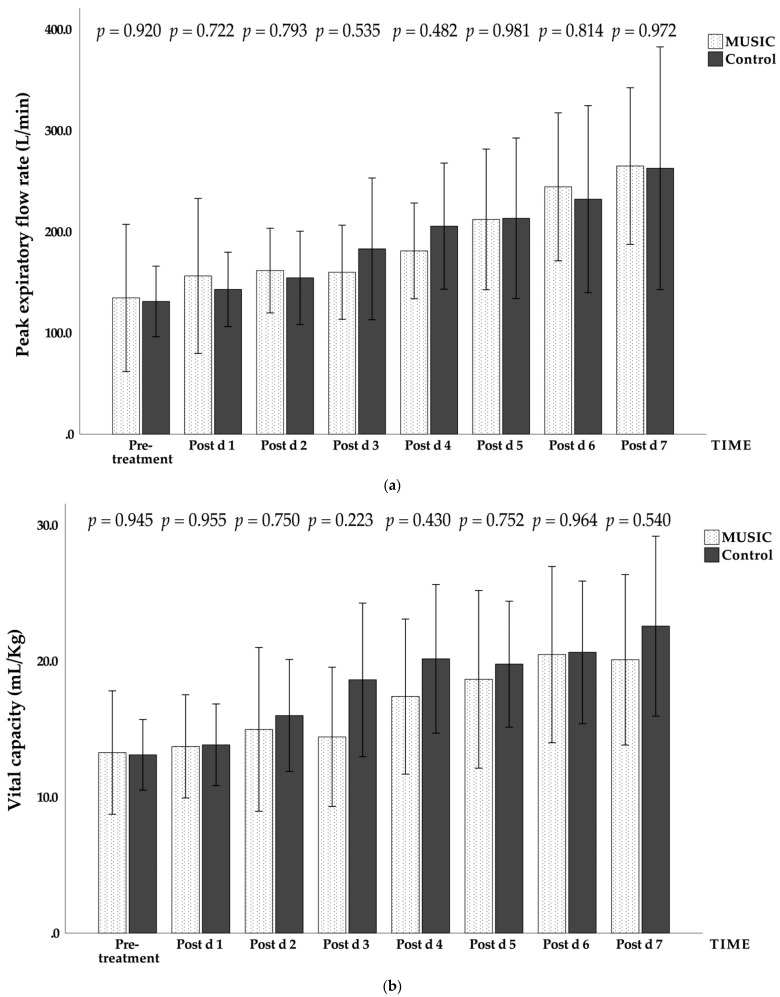
Daily changes in the music group and the control group during the trial. (**a**) Peak expiratory flow rate (PEFR), unit: L/min; (**b**) vital capacity (VC), unit: mL/kg; (**c**) maximal inspiratory pressure (MIP), unit: cmH_2_O; (**d**) maximal expiratory pressure (MEP), unit: cmH_2_O. Mean ± standard error (error bar: 95% CI). Post d, post-treatment day.

**Figure 2 jcm-11-01589-f002:**
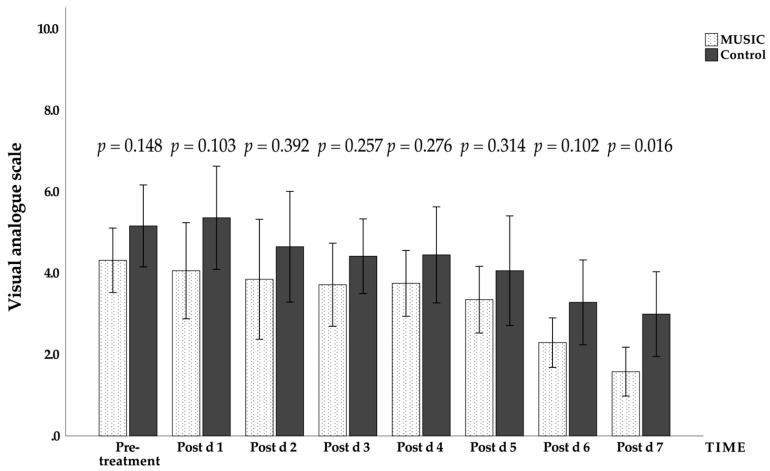
Daily changes in the visual analogue scale for pain in the music group and the control group during the experiment. Mean ± standard error (error bar: 95% CI). Post d, post-treatment day.

**Figure 3 jcm-11-01589-f003:**
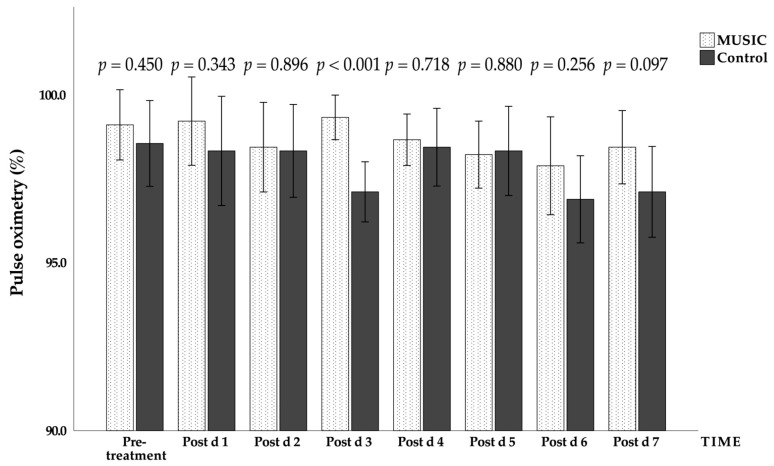
Daily changes in the blood oxygen saturation in the music group and the control group during the experiment. Mean ± standard error (error bar: 95% CI); Post d, post-treatment day.

**Figure 4 jcm-11-01589-f004:**
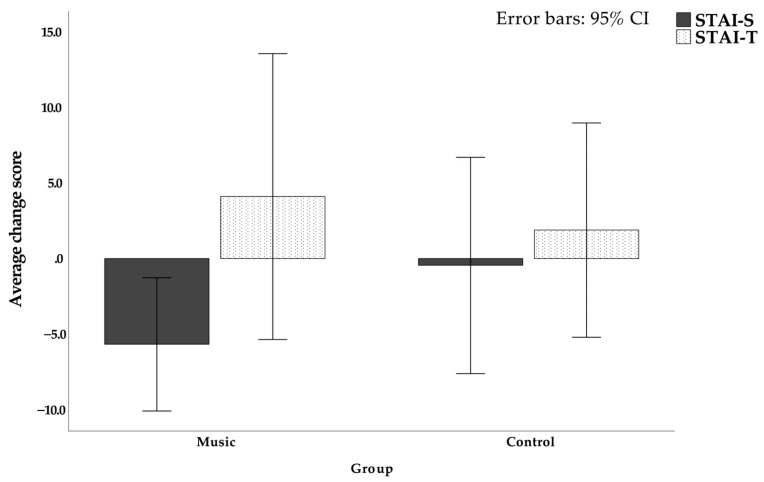
Changes in State-Trait Anxiety Scale before and after surgery in the control group and the music group.

**Table 1 jcm-11-01589-t001:** Classification of patients undergoing cardiothoracic surgery.

	Music Group (*n* = 9)	Control Group (*n* = 9)	*p*
Sex			0.576
Male	6 (66.7%)	8 (88.9%)	
Female	3 (33.3%)	1 (11.1%)	
Type of cardiac disease			
Coronary heart disease (*n* = 7)	3 (33.3%)	4 (44.4%)	
Valvular heart disease (*n* = 6)	2 (22.2%)	4 (44.4%)	
Aortic dissection (*n* = 3)	2 (22.2%)	1 (11.1%)	
Atrial septal defect (*n* = 1)	1 (11.1%)	0 (0.0%)	
Cardiac tumor (*n* = 1)	1 (11.1%)	0 (0.0%)	

**Table 2 jcm-11-01589-t002:** Examination of the differences between the normal distribution and the two groups before the trial.

	Music Group (*n* = 9)	Control Group (*n* = 9)	*W*
Age	58.44 ± 10.06	63.11 ± 11.80	0.380
Height (cm)	164.18 ± 8.15	165.53 ± 9.46	0.749
Weight (kg)	73.79 ± 10.23	70.03 ± 6.40	0.364
Body mass index (kg/m^2^)	27.43 ± 3.81	25.74 ± 3.52	0.343
Heartbeat (bpm)	78 ± 12.40	85 ± 8.54	0.196
Systolic blood pressure (mmHg)	132.11 ± 21.44	125.67 ± 18.26	0.502
Diastolic blood pressure (mmHg)	72.11 ± 17.86	65.22 ± 9.85	0.326
Blood oxygen saturation (%)	96.44 ± 2.96	97.33 ± 2.23	0.483
Vital capacity (mL/kg)	27.82 ± 10.38	23.76 ± 5.46	0.315
Maximal inspiratory pressure (cmH_2_O)	73.33 ± 24.68	56.33 ± 18.24	0.116
Maximal expiratory pressure (cmH_2_O)	92.44 ± 37.38	67.89 ± 16.04	0.089
Peak expiratory flow rate (L/min)	377.78 ± 129.69	371.11 ± 125.44	0.913
State anxiety scale	55.22 ± 11.16	52.22 ± 8.23	0.525
Trait anxiety scale	49.33 ± 14.12	55.78 ± 7.98	0.251

*W*, Wilcoxon signed-rank test.

**Table 3 jcm-11-01589-t003:** Music of model effects in pulmonary function.

Assessment	Group		Time		Group × Time	
	*p*	η^2^	*p*	η^2^	*p*	η^2^
PEFR	0.957	0.000	<0.001	0.831	0.022	0.451
VC	0.567	0.018	<0.001	0.811	0.045	0.417
MIP	0.145	0.105	<0.001	0.755	<0.001	0.673
MEP	0.382	0.041	<0.001	0.844	0.243	0.306

Note: PEFR, peak expiratory flow rate; VC, vital capacity; MIP, maximal inspiratory pressure; MEP, maximal expiratory pressure.

**Table 4 jcm-11-01589-t004:** Effects of music on vital signs and visual analogue scale for pain.

Assessment	Group		Time		Group × Time	
	*p*	η^2^	*p*	η^2^	*p*	η^2^
HR	0.420	0.458	0.066	0.396	0.306	0.285
SBP	0.797	0.004	0.049	0.412	0.115	0.363
DBP	0.756	0.005	0.170	0.335	0.468	0.238
SpO_2_	0.010	0.268	0.010	0.484	0.003	0.523
VAS	0.046	0.178	<0.001	0.739	0.155	0.353

heart rate (HR); systolic blood pressure (SBP); diastolic blood pressure (DBP); pulse oximeter oxygen saturation (SpO_2_); visual analogue scale (VAS) for pain.

**Table 5 jcm-11-01589-t005:** State-Trait Anxiety Inventory: preoperative and postoperative differences.

	Group	Δ_mean_	S.E.	*F*	*p*-Value
STAI-S	Music	−5.667	5.7228	0.956	0.171
	Control	−0.444	9.3020		
STAI-T	Music	4.111	12.3130	0.392	0.671
	Control	1.889	9.2256		

STAI-S, state; STAI-T, trait; ∆_mean_, average change; S.E., standard error.

## Data Availability

The data generated and analyzed in this study are included in this article.

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
