# Peer review of "Effect of Music Intervention on Lung Expansion Exercises after Cardiothoracic Surgery"

_jcm, 2022, doi:10.3390/jcm11061589_

Round 1

Reviewer 1 Report

 Effect of music on lungs expansion after surgery is studied.   Some parameters shows improvements due to music intervention while some are not. There are some queries

  1. Data given in Table -1 are not consistent, like Age, for music group  and control group looks very different. Lungs expansion depend on age. Similarly, blood pressure, lung capacity, maximum expiratory pressure.
  2. Figure 1 is  data font is very small. 

     3. Figure 4 need to explaine in details

Reviewer 2 Report

Thank you for giving me the opportunity to review this manuscript titled “Effect of Music Intervention on Lung Expansion Exercises after Cardiothoracic Surgery” by Dr. Chen and colleagues from the Department of Respiratory Therapy of the Kaohsiung Chang Gung Memorial Hospital, in Kaohsiung, Taiwan.

In this prospective, randomized study performed in their institution the authors assessed the outcome of music intervention on the effectiveness of intermittent positive pressure breathing (IPPB) and in overall respiratory function, anxiety and pain in patients undergoing cardiothoracic surgery. As a cardiothoracic surgeon myself I found the concept and idea quite intriguing because if it is effective it could provide a beneficial intervention to our patients with minimal investment and effort!

The study is quite well designed and conducted. It has a robust methodology with clear inclusion and exclusion criteria, realistic end-points, a sample size calculation was performed prior to commencement, a good randomization and blinding technique and a well performed statistical analysis. The manuscript is also well written in clear and understandable language with only very minor spelling mistakes and well presented.

In reality I only have some minor queries for the authors as I was quite pleased with their work. Specifically:  

  1. The authors omit to mention what type of surgery (cardiac - coronary artery bypass grafting, valve replacement, thoracic - lobectomy, wedge excision) and also what type of excision (sternotomy vs. thoracotomy) was performed. This is very important to know because there are different post-operative chest mechanics between sternotomy and thoracotomy as well as different respiratory functionality following for example a lung excision/lobectomy compared to cardiac surgery. They need to clarify the type of surgery and incision for the reader!  
  2. The authors also do not specify when and for how long the patients heard music. Was it during physiotherapy only or was it throughout the day whenever they wished?
  3. Finally the included figures are quite small and difficult to read. They need a bit of an enlargement and clarification for ease of the readers.

 There are some minor queries I have which I am sure the authors could easily address.

Reviewer 3 Report

Brief Summary: This study is about the effect of natural music intervention lung physiology in cardiothoracic surgery patients. Authors’ aim was to investigate postoperative lung expansion exercises (maximum inspiratory pressure, maximum expiratory pressure, peak expiratory flow rate) as well as anxiety state and trait before and after cardiothoracic surgery. Patients were randomly assigned to receive, or not, natural music intervention. Authors presented that the two groups were almost similar in the pre-trial baseline. In detail

Pulmonary function: no differences in the overall process between groups, while differences were noted longitudinally.

Vital signs: SpO2 was significantly different between groups and longitudinally, but the effect size on heart-rate, and SpO2 was medium.

Pain (VAS scale): VAS was significantly different between groups and daily with time (always lower VAS score in music group)

Anxiety: the music group showed a greater decrease in anxiety state scale, while in anxiety trait scale, changes in both groups were both upward increases.

Authors conclude that nature music intervention has a positive physiologic impact and can reduce post-op pain and anxiety in cardiothoracic surgery patients.

Broad comments: This study deals with an interesting issue, considering that music intervention is easy to implement, cheap, with no adverse events and positive clinical impact (in a wider scope) in elective surgical patients and beyond. The article is well-organized, well-written and easy to follow. Separate sections, Introduction, Methodology etc, are well-developed, as well as the synthesizing of the literature. Authors demonstrate a well-explained sound methodology, though some of the results did not reveal significant associations, such as pulmonary function and vital signs. It is of note the extremely small number of participants, that can affect the results.

Specific comments: In detail,

Abstract: Explanatory, aim, results and conclusion clear.

Introduction: Easy to follow, short, clear aim of the study.

Methodology: The methodology is clearly explained and appropriately structured.

Results: All results are well-written. Some of the results were fruitless, and other did not show sound differences between groups, such as anxiety and pain perception, which makes someone wondering whether the small sample size might play a significant role. Pain and anxiety (patients’ perception) were reduced in music group, which is very promising.

Discussion: The discussion gathers all information nicely of what this study found and added to the literature. Authors critically and in depth discuss their results. Although limitations are nicely addressed, the small sample size is not disclosed, probably because the authors based on G*power the results (page 3, Statistics). I would give a thoughtful consideration on this subject.

Conclusions: Clear

References: please check ref 3, 5, 15 for typos

Thank you for the privilege of reviewing your work.
